# Perceptions around COVID-19 and vaccine hesitancy: A qualitative study in Kaski district, Western Nepal

Preeti Mahato[1,2], Bipin Adhikari[3,4], Sujan Babu Marahatta[5], Susagya Bhusal[5], Kshitij Kunwar[6], Rajesh Kumar Yadav[7], Sushila Baral[8], Anisha Adhikari[9], Edwin van Teijlingen[2]*

**1** Department of Health Studies, School of Life Sciences and the Environment, Royal Holloway University of London, Egham, United Kingdom, **2** Centre for Midwifery, Maternal & Perinatal Health, Bournemouth University, Bournemouth, United Kingdom, **3** Mahidol-Oxford Tropical Medicine Research Unit, Faculty of Tropical Medicine, Mahidol University, Bangkok, Thailand, **4** Nuffield Department of Medicine, Centre for Tropical Medicine & Global Health, University of Oxford, Oxford, United Kingdom, **5** Manmohan Memorial Institute of Health Sciences, Kathmandu, Nepal, **6** Central Department of Public Health, Tribhuwan University, Kirtipur, Nepal, **7** One Heart World Wide, Kathmandu, Nepal, **8** Provincial Health Training Centre, Gandaki Province, Nepal, **9** School of Health and Allied Sciences, Pokhara University, Pokhara, Nepal

* evteijlingen@bournemouth.ac.uk

**Data Availability Statement:** In line with good practice for reporting qualitative research excerpts of the interview transcripts relevant to the study are

## Abstract

Burgeoning morbidity and mortality due to COVID-19 pandemic including the peaks in outbreaks due to different variants have attracted global attention. Although the development and rolling out of vaccines have been impressive, low- and middle-income countries suffer from a double burden: (1) lack of adequate vaccines; and (2) low vaccine uptake (vaccine hesitancy). The main objective of this study was to explore perceptions around COVID-19 and vaccine hesitancy among urban and rural population in Western Nepal. A qualitative study was conducted in six urban wards of Pokhara municipality and four rural municipalities in Kaski district of Nepal. A semi-structured interview guide was used to interview participants who were selected purposively to explore the perceived burden of COVID-19 pandemic, roles, and contributions of vaccine. Nineteen interviews were conducted by telephone following a government recommendation to avoid face-to-face meetings. Audio-recorded interviews were thematically analysed after transcription and translation into English. COVID-19 is a major (public) health concern and affects people at an individual, societal and national level. People dreaded its health hazards and consequences and seemed to be compliant with public health measures such as maintaining social distance, wearing masks and maintaining hygiene. Vaccine was considered to be a major intervention to fight the pandemic, nonetheless, the rationale and benefits of vaccines were blemished by the perceived lack of the vaccine's effectiveness, duration of protection, and its potential side-events. Expedited development of vaccine was embraced with suspicion that vaccine may have incurred compromise in quality. Science and rationale behind vaccine were smeared by misinformation and clearly counteracting the misinformation were deemed critical. Providing information about vaccines through government entities (who are trusted) and respected individuals may engender trust and uptake of vaccine. Fighting off misinformation of COVID-19 is critical to curb the course of pandemic. Increased attention towards

cited in this paper. Unfortunately, due to the wording of the promise of confidentiality to all interviewees in Nepal the data are only available for further research by external researchers under supervision of our researchers, i.e. the only people with access to the data files. The authors are happy to discuss further analysis of the data with other researchers if the research question is appropriate, but we can't share the qualitative data.

**Funding:** This small-scale study was funded by Bournemouth University who supported it with 1,460 pounds. The grant holders were PM & EvT.

**Competing interests:** The authors declare there are no competing interests.

monitoring and investing in legitimacy of information and offering information through trusted sources can help improve the vaccine coverage.

## Introduction

The Coronavirus Disease-2019 (COVID-19) pandemic has become a major public health threat since its outset in 2019 in China [1, 2]. The rapid spread and infectiousness of COVID-19 from China to rest of the world prompted the World Health Organization (WHO) to declare it as a pandemic in March 2020 [3]. Nepal, neighbouring China, reported its first case on January 13, 2020, in a student returning from Wuhan [4]. Notwithstanding the first case of COVID-19, with more than 1600km of open border with its southern neighbour, the majority of Nepal's infectious disease burden mirrors India's epidemiology, including in malaria [5], HIV (Human Immunodeficiency Virus), dengue [6] and more recently COVID-19 [7]. The COVID-19 disease burden and its management challenged Nepal's health system with its resource constraints, and continues to pose a threat if preventive measures, such as maintaining physical distance, wearing masks, avoiding crowds, and mass vaccination are not being implemented [8].

Effectiveness of vaccines depend on vaccine availability and uptake, and vaccine hesitancy, the latter can range from simple indecisiveness/doubt to outright anti-vaccination beliefs [9–11]. Vaccine hesitancy was listed as one of the top ten threats to global health in 2019 by the WHO and is a critical barrier in preventing vaccine preventable diseases [12]. Vaccine hesitancy has gained increasing attention over the last few decades and is deemed to play a central stage during this pandemic [9, 10]. Undoubtedly, vaccine hesitancy has become a bigger problem than anticipated, particularly how development of vaccine was perceived to be the end of the game for the current pandemic. Unfortunately, vaccine hesitancy has jeopardized the optimism offered by development of vaccine and continues to ravage the population because of emergence of new variants [13]. Vaccine hesitancy is an important barrier in achieving high vaccination coverage and has been reported among various socio-demographics from educated populations [14], who failed to appreciate the science of vaccines, to refugees in the USA [15]. Despite the volume of literature around vaccine hesitancy, it has remained to be an elusive topic, particularly because of its complexity and the myriad factors affecting it. Some of the prominent reasons centre around the concerns of safety, potential adverse effects, and disconcerting rumours around vaccine's impact on fertility and pregnancy [16]. Understanding how vaccine is perceived in a particular context and factors affecting its acceptance (or its absence) can offer a window to a potential success of vaccine roll out.

Nepal tackled COVID-19 pandemic amidst the ongoing transformation in health system and corresponding challenges [8, 17]. Nepal detected a total of 979, 607 cases, and almost 12,000 lost lives due to COVID-19. Almost 68% of population has received two doses of COVID-19 vaccines with 9% partially vaccinated, against the backdrop only 17.8% in all low-and-middle-income countries (LMICs) [18]. However, one in third of Nepal's population is still unvaccinated, partly due to people's perceptions around COVID-19 and vaccine hesitancy. The main objective of this study was to explore perceptions around COVID-19 and vaccine hesitancy among urban and rural population in Kaski district of Western Nepal.

## Materials and methods

### Study settings

Kaski district was purposively selected as it has a good mixture of rural and urban areas (including the second largest city in the country Pokhara metropolitan city with 33 wards, and

**Table 1. Socio-demographics of participants in the study.**

| N°. | Place | Occupation | Age | Gender | Religion | Status COVID Vaccine | Education level |
|---|---|---|---|---|---|---|---|
| 1 | Rural | Nursing Instructor | 29 | F | Hindu | Vaccinated | Higher |
| 2 | Urban | Homemaker | 28 | F | Hindu | Unvaccinated | Higher |
| 3 | Urban | Admin-Finance Officer | 32 | F | Hindu | Unvaccinated | Higher |
| 4 | Urban | Business and job | 30 | M | Hindu | Unvaccinated | Higher |
| 5 | Urban | Health worker | 29 | M | Hindu | Unvaccinated | Higher |
| 6 | Urban | Engineer | 31 | F | Muslim | Unvaccinated | Higher |
| 7 | Urban | Business | 57 | M | Hindu | Unvaccinated | Secondary |
| 8 | Urban | Homemaker | 56 | F | Hindu | Unvaccinated | Uneducated |
| 9 | Urban | Homemaker | 61 | F | Hindu | Unvaccinated | Primary |
| 10 | Urban | Health worker | 21 | F | Hindu | Vaccinated | Secondary |
| 11 | Rural | Health worker | 42 | M | Hindu | Vaccinated | Secondary |
| 12 | Urban | Retired Indian Army | 74 | M | Hindu | Unvaccinated | Primary |
| 13 | Rural | Health worker | 26 | M | Hindu | Vaccinated | Higher |
| 14 | Rural | Health worker | 26 | F | Hindu | Vaccinated | Higher |
| 15 | Urban | Public Health Officer | 26 | F | Hindu | Vaccinated | Higher |
| 16 | Rural | Retired Army | 63 | M | Buddhist | Vaccinated | Secondary |
| 17 | Rural | Retired Teacher | 63 | M | Hindu | Unvaccinated | Secondary |
| 18 | Rural | Retired Civil Servant | 59 | M | Hindu | Vaccinated | Higher |
| 19 | Rural | Social Service Activist | 32 | F | Hindu | Vaccinated | Secondary |

four rural municipalities (Rupa, Madi, Machhapuchre and Annapurna). From the city, six wards were selected to represent urban areas and four from the rural areas. A ward is an administrative unit comprising around 12 thousand people. A total of 11 urban and eight rural participants were selected in this study (Table 1).

For the recruitment of participants, we asked the health section of Pokhara Metropolitan City to provide the contact details of health facility managers of selected wards. The latter were briefed about study and asked to provide details of potential interviewees, who could be purposively selected based on the diversity and who may offer us valuable insights in line with the tenets of qualitative research [19]. Potential interviewees were then contacted and briefed about the study and asked if they were willing to participate. Because of COVID-19, all interviews were conducted remotely through mobile phones. Interviews were audio-recorded in mobile phones after obtaining their verbal consent for the interview and audio-recording.

## Study design

This is a qualitative study conducted with purposively selected participants from Kaski district in western Nepal and follows a standard consolidated criteria for reporting qualitative studies (S1 Checklist) guideline [20]. Members of the research team have conducted several studies and worked in health sector in the region and have extensive knowledge about health services provided in the region.

## Study guide and data collection

A semi-structured interview guide was developed to address the research question. The interview guide was prepared by the first author with inputs from second and the last author. It was first piloted among two local participants and was amended based on their feedback. Additional information regarding the ethical, cultural, and scientific considerations specific to

inclusivity in global research is included in the S3 File. The interview guide focused on barriers and facilitators of vaccine hesitancy and its uptake (S2 File)

Nineteen interviews were conducted through telephone in May 2021 because of the restrictions imposed due to the pandemic. No repeat interviews were conducted. The number of interviewees was based on reaching data saturation whereby subsequent interviews did not yield new information [21]. None of those contacted refused to be interviewed.

The data were analysed using inductive thematic analysis technique. Interviews were conducted in Nepali language at a convenient place either home or workplace using the interview guide which was translated in Nepali by RKY and SB. All interviews were audio recorded by lead interviewers, one male (RKY) and two female (SB and AA). Field notes were also taken during the interviews. The recorded interviews were transcribed and translated into English by KK and SBh. Transcripts were cross-checked by PM. The transcripts of the interviews were not returned to participants for comments.

All research assistants who conducted the interviews (RKY, SB, AA) are postgraduate students at Pokhara University. Two research assistants (RKY and SB) had already been trained in interview techniques as they had previous conducted qualitative research. The third research assistant (AA) received training in interviewing from the first author (PM), who has almost ten years of experience in conducting qualitative research. The first author completed her undergraduate degree and worked for three years in Nepal including two years as a government Public Health Officer.

### Data analysis

Thematic analysis of the transcripts was conducted by the first author and samples from the transcripts were cross checked by KK and SBh. An inductive approach to thematic analysis was undertaken to ensure all themes were identified. No software was used for the data analysis. Final themes were discussed among the authors and were categorised into main themes and sub themes based on the relevance to the research question, and excerpts of the transcripts relevant to the study are cited in this paper.

### Ethics

Ethical approval was sought and obtained in Nepal from Manmohan Memorial Institute of Health Sciences ethical board (MMIHS-IRC 585). The participants were contacted and briefed about research by telephone. Verbal consent was taken over the telephone before conducting interviews. Participants were informed that confidentiality would be maintained during the research. They were also made aware that their participation was voluntary and could withdraw their participation at any point without any consequences.

## Results

There were 19 interviewees, ten women and nine men. The interviews lasted between 12 and 29 minutes. The average age of participants was 41.3 years, and nine were vaccinated and 10 unvaccinated, 11 lived in urban areas while the rest lived in rural areas, the majority (n = 17) of them were Hindu and nine were educated to university level (undergraduate degree and above).

The figure depicting the themes and sub-themes is shown in Fig 1.

### Experience, perceptions, and impacts related to COVID-19

**Fear, contagiousness, and impacts.** Most participants perceived the pandemic amounting to protracted fear and exhaustion among themselves and their family and peers. Most

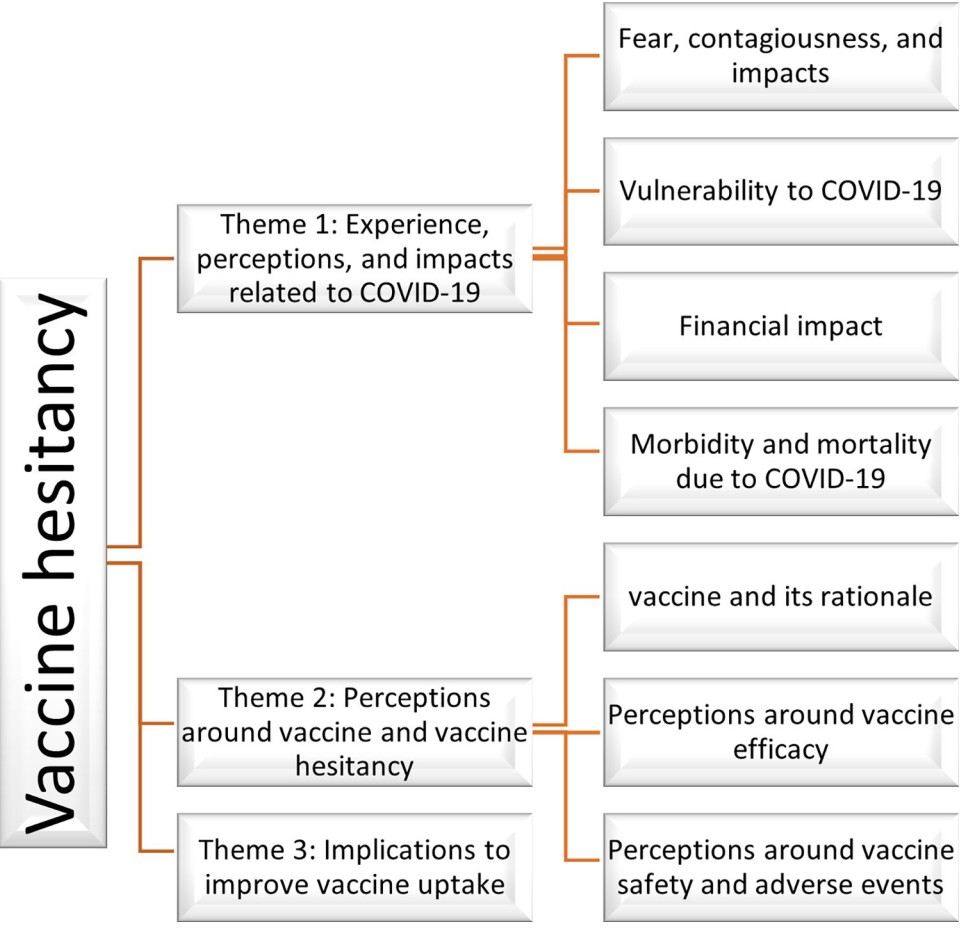

**Fig 1. Thematic analysis demonstrating hierarchy of themes.**

participants felt COVID-19's second wave was more dreadful than the first one. This fear was exacerbated by their job and the exposure, implying the high contagiousness and transmissibility of COVID-19. For instance, one interviewee feared when potential exposure (due to touch or when being near to COVID patients) could infect them.

> *"When I work in hospital, I feel it very fearful to touch the patients even when I am putting on a PPE." P10, 21 years, urban female*

Although this fear was linked to their exposure and experience how they and their peers and family contracted the disease, one interviewee thought that the pandemic was the result of a conspiracy. The pandemic was perceived to be some form of war and COVID was its biological weapon.

> *". . . it might be a form of the third world war . . . that's what I think. This might be a biological war started by a country, this might be a disease run by the smugglers, that's what I feel" P17, 63 year, rural male*

Upon further probing, interviewees explained that they were mostly worried of losing their lives and linked such fears to their lived experience of seeing peers and family dying of

COVID. They explained that this resulted in constant fear and psychological stress, particularly since these fears were intensified by media (television and social media).

*"The pandemic has not only affected the health of the people but has also impacted their mental and economic status." P4, 30 years urban male*

Psychological impacts including fear was further exacerbated by the imposed lockdown which restricted their outdoor lifestyles.

*"While we youth have the internet to spend our time on, the children and the elderly have been irritated by staying inside home for long time." P6, 31 years, urban female*

Interviewees also reported having suffered from COVID-19 related stigma and discrimination One of the health workers had experienced discrimination due to being a health care worker. Apparently, people who discriminated because they assumed that health care worker could contaminate them with the disease.

*"I had to experience social discrimination when I got infected from COVID-19." P14, 26 years, rural female*

**Vulnerability to COVID-19.**   People were generally aware of the causative agent, its origin (as China), and how it transmits from person to person. Most stated that immunity power was important and were well-versed in some of the high-risk conditions such as chronic conditions due to diabetes, old age, cancer, and concomitant diseases.

*Although people like us who have good immunity power might not suffer from the infection, we can transmit it to the people with poor immunity conditions such as elderly." P10, 21 years urban female*

Many also stressed the value of preventing from the pandemic by adopting public health measures rather than being risk to the self and others. The second wave was particularly considered to be a lesson for them as many of their known contacts were either sick or had died. Nonetheless, there were rumours that even vaccinated people had severe COVID and died because of it. Apparently, such perceived severity incited people to be more careful and adopt preventive measures.

## Financial impact

Most interviewees spoke about the impact of the pandemic on their daily work, opportunities and jobs ultimately affecting their financial stability. Increase in unemployment, loss of jobs and having to pay for treatment of COVID impacted their lives.

*". . ..and they [COVID affected persons] are not able to present themselves in the place of their work and that has caused a high impact on their economic stability" P13, 26 years, rural male*

Interviewees also expressed the aftereffects of pandemic due to public health measures such as imposition of nationwide lockdown that affected population in multiple ways. The impact was reported to be affecting most of the poorest, those on daily wages, for who a day of lock down affected their income directly. Daily living was further affected by the increase in food

prices due to supply-chain disruption. Many also explained about lockdown's impact on non-COVID health services including the adverse impacts on some of the chronic conditions and disruption in availability of regular medicines.

## Morbidity and mortality due to COVID-19

Many interviewees shared stories of morbidity and mortality among friends, peers, and family. Although most interviewees were not infected with COVID-19, three spoke about coping with the infection and two about losing their relatives to COVID-19. Among those who were infected, their COVID-19 symptoms included: cough, fever, body ache, and weaknesses. One interviewee shared his difficult recovery on an intensive care unit (ICU). The difficulties in finding a hospital and a bed for COVID patient was further compounded by limited availability on hospital beds and the rise in costs for such bed. It was deemed simply unaffordable for poor patients.

## Perceptions around vaccine and vaccine hesitancy

**Vaccine and its rationale.**   Most intervieweess explained the role of vaccine in curbing the pandemic including some clues about how it may promote the immunity against corona virus. These interviewees were firm in their belief that vaccine was necessary to counteract the pandemic.

*"I don't think the pandemic will end until a vaccine effective against the infection will be there." P6, 31 years, urban female*

Although the vaccine was seen as critical in the fight against the pandemic, interviewees had mixed opinions due to conflicting news and messages from various informal sites and social media. They often resorted to recommendations by WHO for vaccine, although the interpretation of information were often blurred. Clearly, the amount of information and comprehensibility of the science behind the vaccine was another major issue for most:

*"The vaccine that is not recommended by WHO should not be used." P4, 30 years, urban male*

Others were clearly confused about how vaccines are meant to work:

*"I am in confusion whether to get vaccinated because they say that vaccinating means inject the virus." P8, 56 years, urban female*

Few interviewees were uncertain about getting vaccinated in the future, as they had received conflicting information from various sources. Such information affected their decision, resulted in alternative theories. One interviewee believed that good dietary habits alone could build strong immunity to fight off the COVID-19 infection.

*"If we maintain good dietary habits, promote our immunity, I don't see the need for getting vaccinated against COVID-19." P7, 57 years, urban male*

Others believed that those who were naturally infected by the COVID developed antibody against the virus and would not need the vaccine. Vaccine was perceived to be an experiment. Few thought that rolling out of vaccine was to promote the psychological confidence and mental health of people rather than prevent the infection.

*"The vaccine helps to boost the mental health of the people to which the physical well-being is also linked." P4, 30 years, urban male*

Interviewees also reported on mixed information about herbs. Some share that their neighbours and relatives believed that herbal medicines were effective against the COVID-19 infection. Some believed that herbal medicines were an alternative to the allopathic medicines, and even warned not to visit doctors.

*". . .some say that one should not visit the doctor, but the herbs will cure the disease, that may be also right, but one should get vaccinated, I believe in vaccine." P12, 74 years, urban male*

**Perceptions around vaccine efficacy.** Few interviewees showed concerns around the effectiveness of the vaccine, particularly as thye heard that some vaccinated people were sick due to COVID-19 and this led them to believe that that vaccine alone was not adequate to protect against the disease. Furthermore, some thought that the vaccine was an adjunctive to other protective measures.

*"Although vaccines have been produced for COVID, there is no proof of the mechanism of a certain vaccine working to prevent COVID. People who had been immunized have also been infected with the disease again. So, I do not think that it'll stop the pandemic, but it will help prevent the infection". P1, 29 years, rural female*

Some mentioned the heterogeneity in effectiveness of available vaccines. Availability of various vaccines was taken as differences in effectiveness. Among various issues related to vaccines, many were more concerned about the duration of protection offered by vaccines, particularly amidst the emergence of new variants. These concerns had prompted them to get vaccinated, and they thought the vaccine was critical and needed further improving.

*"I am in confusion about the effectiveness of the vaccine because there are no long-term studies on it. . ...As the death rate because of COVID-19 is rising day by day, . . . . . . . .in order to protect myself form the virus I think it might be beneficial to be vaccinated." P2, 28 years, urban female*

*"The only way to tackle this disease is vaccine, there is nothing else, but I think that there need to be some improvement on the previous vaccine, I mean one should not be infected after getting vaccinated." P3, 32 years urban female*

Nonetheless, in the interviews some of the rumours and negative perceptions around vaccine such as 'vaccine could impair reproduction and fertilization' were shared. Interviewees showed concerns that echoed the global news related to vaccine, particularly related to vaccines and blood clots.

*". . .in UK the vaccine has caused clotting of blood and some say that it affects the reproduction and fertilization potential." P1, 29 years, rural female*

Interviewees expressed concerns around the potential compromises in the quality of vaccine because of the short period in which these vaccines were developed, which was perceived to be affecting their efficiency against the new variants. One person stated that people had

diminished enthusiasm for the vaccine due to uncertainty surrounding the availability of second dose, leaving them vulnerable to infection.

*"Most of the people do not vaccinate themselves because they are not sure if and when they are going to get the second dosage. One of my friends did not vaccinate because of the same reason, the vaccine was there for health workers, but they were not sure if they would also receive a second dose." P1, 29 years, rural female*

**Perceptions around vaccine safety and adverse events.** Several interviewees shared other people's experience of adverse events after taking the vaccine which included fever, vomiting, headache, body pain, and insomnia. One man shared his experience of having severe side-effects and being bed-ridden for over a week.

*"I am worried because I have heard that there are severe side effects of the vaccine like fever, body pain, vomiting and diarrhoea." P5, 29 years, urban male*

Few interviewees heard among their family, friends and neighbourhood about the side-effects of taking vaccine, some had heard of death occurring due to vaccine. Some also highlighted vested interests in the roll-out of the vaccine such as political incentives.

*"As they said the vaccine has some side effects, even in some cases people face more difficulty, because of the severity even the rumour of death was also heard. Seeing this I get scared." P2, 28 years, urban female*

**Implications to improve vaccine uptake.** Some interviewees offered suggestions on how to improve vaccine uptake using suitable strategies. Most of these suggestions were around how to offer correct information, counteract negative information about vaccines and reaching as many people as possible. Many suggested to increase the awareness related to vaccine using media by government of Nepal (rather than other informal sources such as social medias).

*"People tend to focus more on the negative side of the vaccine rather than the positive ones so the government should inform people about vaccine clearly. . .. The government should tell people about why one should get vaccinated " P14, 26 years, rural female*

*". . .there are many people in the social media these days, maybe people are not getting vaccinated because they have heard negative things about the vaccine. . ." P13, 26 years, rural male*

Intervieweess also suggested the need to provide appropriate information about vaccines, their importance, and adverse effects, especially to mitigate potential rumours and confusions related to need of vaccine for patients with chronic conditions and pregnant women or lactating mothers. They also suggested highlighting a brief history of vaccine, and promoting success stories in counteracting diseases.

*"Any drugs or vaccine has got its side-effects, we need to better inform people about it." P10, 21 years, urban female*

Some went further than providing information, and stressed the importance of educating the general population:

*". . .there is a need to spread awareness on the society, we have to educate people. I have heard a rumor that those who get vaccinated can become infertile. So, we have to provide information to people and make them aware, educate them, make them literate. . ." P7,57 years, urban male*

Interviewees also suggested the need to counter misinformation by utilizing health workers as information providers. In addition, respected and popular people could be asked to support the dissemination of information. Sharing of lived experience by health workers and government officials were deemed to be more convincing than media alone.

*"We (health workers) should come out from our side and provide information to people about what is the purpose of vaccine and how it works. If we are able to provide this information to people, I think they will be motivated to get vaccinated."*

P13, 26 years, rural male

## Discussion

Most concerns were raised around COVID-19 as a major infection that could affect health, livelihood and increase the vulnerability. Vulnerabilities towards COVID-19 were seen as high and persistent despite adopting public health measures such as maintaining distance and wearing mask. Vaccine was deemed to be a critical and unique intervention to counteract the adverse consequences of COVID-19 [22]. Nonetheless, fear and suspicion towards the effectiveness of vaccine was prominent and were associated with the concerns around type of vaccine, process of its development, doses, protection-duration offered by the vaccine, and perceived/potential adverse effects of vaccine. Fear and suspicion towards vaccine resembled a global phenomenon of vaccine hesitancy and requires concerted intervention to promote the vaccine uptake and coverage.

### Experience, perceptions, and impacts related to COVID-19

Fear and apprehension due to COVID-19 had an impact on their daily living, social interaction, and psychological well-being and echoes with the existing literature [23–25]. Fear surrounding COVID-19 was associated with their vulnerability due to the highly contagious nature of the disease, lack of medicine, and (full doses) of vaccines, and such fears, and stigma were higher among the health workers [8, 26]. Undoubtedly, fears were further exacerbated by the shared anecdotes from their peers, neighbours and family members, how they journeyed through disease including the loss of loved ones [27]. The share of their fears was also clearly acknowledged by the constraints in Nepal's health system, particularly overburdened health services, inadequate number of ICUs, and potential catastrophe due to combination of these factors [8, 23, 28]. Although fear surrounding COVID-19 was a global phenomenon, such fears are protracted, and population feel vulnerable in low- and middle-income countries due to constraints in health system. Fear of COVID-19, specifically high perceived risk was an important driver of intention to get vaccinated in Bangladesh [29]. Such constraints have been recognized as barriers to access and uptake of health services in Nepal in various health conditions [30, 31]. Impediments in health services were prominent due to COVID-19 to an extent

that population was deprived of primary health care services, including quintessential maternal and child health care [23, 32].

## Perceptions around vaccine and vaccine hesitancy

Most interviewees saw vaccines as pivotal in curbing the pandemic. Nonetheless, their convictions around vaccine were not devoid of false rumours, and beliefs. Much of their concerns were also associated with the safety, effectiveness and the adverse events due to vaccines [22]. Interviewees demonstrated a wide extent of suspicion towards the rationale of vaccine, its effectiveness, and the alternatives. A lot of these concerns are rooted to the incomprehensibility of science, mechanism how vaccine works, and the rumours built around partial foundations of knowledge. For instance, some explained their apprehension to take vaccine as they have heard that these vaccines include the virus and therefore vaccinating meant injecting the virus. This demonstrates incomplete understanding of science and also implies how trust on medical interventions can and should short-circuit the uptake [13]. One of the prominent findings in this study is the impact of various information they have heard or received from different sources, a lot of which apparently were convincing and rational. For instance, maintaining good dietary habits, and exercise does promote immunity, however, overstretching their impacts to underestimate the need of vaccines can have dreadful consequences. These findings echoed with studies from neighbouring countries. In a nationwide study in India, more than one third of participants were either unsure or unwilling to take COVID-19 vaccine when available. The majority of participants had concerns regarding side effects, effectiveness and rapid development of the vaccines (short duration to produce the COVID-19 vaccines) [33]. In a nation-wide study in Bangladesh, vaccine hesitancy was associated with bearing negative attitude, mistrust and conspiracy beliefs towards COVID-19 vaccines [34].

This pandemic was also a perfect storm for uncertainty and thus was fertile to alternative sources of information including treatment. Promotion of herbs/herbal substance were widespread and had adverse consequences to allopathic medications including vaccines. One of the prominent examples that echoed such practice was the false claims around corona cure by a revered Hindu Yoga guru in India [35].

Suspicion towards an effectiveness of vaccine also originated from anecdotal experience of hearing or seeing some of the vaccinated people who caught COVID-19—breakthrough infections [36]. Seemingly, such anecdotes contravened their beliefs that 'vaccine should be omnipotent'. Some thought that there was little evidence to support vaccine was working. While on one hand, people expect a perfect vaccine, on the other hand, potential role of vaccines are marred by false rumours that the vaccine could impair fertility [37]. Short span of time to produce the vaccine was an extraordinary effort by 'operation warp speed', unfortunately, such efforts were also alleged to have compromise the quality of vaccine [38]. Another established reason surrounding vaccine hesitancy are adverse events associated with the vaccine. Exaggeration of adverse events, and shortcomings that are characteristics of any intervention apparently predominate the narratives to jeopardize the impression of vaccine. Accentuation of negativity and adverse events probably flares-up more than expected because of the human nature of being receptive and interested to unusual circumstances, impact and news [39, 40].

## Implications to improve vaccine uptake

Most people in this study offered recommendations around counteracting misinformation— seen as major problem in creating confusion among population [41–43]. One of the strategies offered was how governments, and people who are respected (e.g. health workers) could be the source of information. Rationale for having government or popular/respected figures

spreading the information lies in the legitimacy of the information and trust inherent in the institution and their profession [44]. Similar measures of using influential leaders and frontline health workers as a source of information was also recommended by a study in India [45], and of Nepalese people living in the UK [46]. Increasing the positive information around vaccine, its rationale and benefits are ever more important, especially as WHO warns that we are fighting off the parallel pandemic due to misinformation referred as 'infodemic' [43]. Indeed, the legitimacy and trustworthiness of information source is paramount and is pivotal to engender trust towards the information and thus vaccine [47]. Briefly, findings from this study can be implicated towards tackling the misinformation (Box 1).

---

### Box 1. Recommendation for improving uptake of vaccine

#### Recommendations regarding providing correct information

• Providing accurate information about vaccines, its importance and adverse effects through formal outlets such as national television channels, radio and newspapers.

• Highlight informative narratives including history and benefits of vaccines.

• Provide success stories of vaccine uptake, its benefits from the region and outside.

• Keeping easily accessible and free sources for fact check such as 'Our World in Data, and 'Worldometers.'

#### Recommendations on counteracting misinformation

• Remind population that not all information is of equal quality, the legitimacy of information.

• Promote health workers as a source of information

• Set-up a mechanism to rapidly react to potential rumors and confusion related to vaccine.

• Government to be prepared to counteract rumours and misinformation early.

• Involving influential figures (e.g. religious leaders, authorities, celebrities) to counteract rumours and misinformation.

---

### Strengths and limitations of the study

The study has several strengths but also limitations. We interviewed a diversity of participants for example health workers, lay population that constituted a mix of vaccinated and unvaccinated, urban and rural residents, from various religions and socio-demographic backgrounds. Study could have suffered from social desirability bias. Since this research was conducted in a hilly district of Nepal, it may not be generalisable to the entire population of Nepal. Future studies can expand and highlight on the correlates of vaccine hesitancy through quantitative surveys, expanding it to the region and the nation. Because of the qualitative nature of the study, prevalence of vaccine uptake (both partial and complete) was beyond the scope of the study. Exploring the prevalence of vaccine uptake and factors affecting could have added to the current level of knowledge.

## Conclusion

This study offers important insights in the spectrum of perceived health concerns around COVID-19, its impact, and potential solutions to fight off the pandemic. Apart from adhering to public health measures, the vaccine was deemed the most essential tool to fight the disease. Nonetheless, the rationale and benefits of vaccines were tainted by misinformation, rumours and narratives around its safety and effectiveness. The glimpses of vaccine hesitancy among the respondents and its potential reasons offer Public Health opportunities to counteract the potential catastrophe due to poor vaccine coverage. Monitoring the misinformation, including intensifying the legitimate information by government and through the involvement of trusted/popular figures can promote the trust in the information, and vaccine ultimately promoting uptake.

## Supporting information

**S1 Checklist. Consolidated criteria for reporting qualitative studies (COREQ).**
(DOCX)

**S1 File. Interview guide in Nepali.**
(DOCX)

**S2 File. Interview guide in English.**
(DOCX)

**S3 File. PLOS' questionnaire on inclusivity in global research.**
(DOCX)

## Acknowledgments

We would like to thank all participants in this study. Also, our sincere thanks to personnel at Pokhara Metropolitan City, health facility managers and facility in charge of the respective wards who helped with recruitment of participants. We also thank the reviewers of this paper for their insightful and positive comments and suggestions.

## Author Contributions

**Conceptualization:** Preeti Mahato, Edwin van Teijlingen.

**Data curation:** Preeti Mahato, Bipin Adhikari, Susagya Bhusal, Kshitij Kunwar, Rajesh Kumar Yadav, Sushila Baral, Anisha Adhikari.

**Formal analysis:** Preeti Mahato, Bipin Adhikari, Susagya Bhusal, Kshitij Kunwar, Sushila Baral.

**Funding acquisition:** Preeti Mahato, Edwin van Teijlingen.

**Investigation:** Preeti Mahato, Susagya Bhusal, Kshitij Kunwar, Rajesh Kumar Yadav, Sushila Baral, Anisha Adhikari.

**Methodology:** Preeti Mahato, Bipin Adhikari, Sujan Babu Marahatta, Edwin van Teijlingen.

**Project administration:** Sujan Babu Marahatta.

**Resources:** Edwin van Teijlingen.

**Supervision:** Bipin Adhikari, Sujan Babu Marahatta, Edwin van Teijlingen.

**Validation:** Bipin Adhikari, Edwin van Teijlingen.

**Visualization:** Bipin Adhikari, Edwin van Teijlingen.

**Writing – original draft:** Preeti Mahato.

**Writing – review & editing:** Preeti Mahato, Bipin Adhikari, Sujan Babu Marahatta, Susagya Bhusal, Kshitij Kunwar, Rajesh Kumar Yadav, Sushila Baral, Anisha Adhikari, Edwin van Teijlingen.

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
