## [Decision Letter · Decision Letter 0]

13 May 2022

PGPH-D-22-00044

Perceptions around COVID-19 and vaccine hesitancy: A qualitative study in Kaski district, Western Nepal

Dear Dr. van Teijlingen,

Thank you for submitting your manuscript to PLOS Global Public Health. After careful consideration, we feel that it has merit but does not fully meet PLOS Global Public Health’s publication criteria as it currently stands. Therefore, we invite you to submit a revised version of the manuscript that addresses the points raised during the review process.

EDITOR COMMENT: Thank you for submitting this manuscript for consideration. I am sorry for the delay in the journal returning a decision. I would like to invite you to revise your manuscript and submit for reconsideration. Please ensure to address the following points:

It is not clearly explained who the participants were. It is stated that "Details of participants were obtained from health facility mangers" but it's not clear how these managers selected them. Please provide clear inclusion criteria and describe how participants were identified. Please remove duplication of material regarding identification of participants from the bottom of pages 9 and 10.You state that "The number of participants in this study was deemed adequate based on the principles of data saturation whereby subsequent interviews did not yield new information". Do you mean that you conducted interviews until you reached saturation?Please provide dates when interviews took place. This is crucial context given the changing dynamics of the pandemic.Please shorten and refocus introduction. The first paragraph of the section "Role of vaccines in preventing morbidity and mortality" can be removed. Keep material relevant to the research topic. For example we don't need to know too much detail about how Nepal's health system is structured, but it would be useful to know about the course of pandemic - cases/ deaths relative to other countries, vaccination rate - up to the point when interviews took place.What language/s were interviews conducted in?Why was Kaski district selected?Please reduce the existing discussion and add material on how your findings compare to findings from other countries?You should include 1-2 paragraphs regarding study limitations. e.g. how generalisable are the findings to Nepal's population?Please provide the interview guide/schedule. If it fits in a table/ box include in main manuscript, otherwise as supplemental material.Please add a box/ table with a summary of recommendations for policy makers/ health services on how to increase vaccine uptake.First/last/corresponding authors are affiliated in UK. You may want to mention that first author completed undergraduate studies in Nepal.

We look forward to receiving your revised manuscript.

Kind regards,

Dmitri Nepogodiev

Guest Editor

Journal Requirements:

1. Please include a complete copy of PLOS’ questionnaire on inclusivity in global research in your revised manuscript. Our policy for research in this area aims to improve transparency in the reporting of research performed outside of researchers’ own country or community. The policy applies to researchers who have travelled to a different country to conduct research, research with Indigenous populations or their lands, and research on cultural artefacts. The questionnaire can also be requested at the journal’s discretion for any other submissions, even if these conditions are not met.  Please find more information on the policy and a link to download a blank copy of the questionnaire here: https://journals.plos.org/plosone/s/best-practices-in-research-reporting. Please upload a completed version of your questionnaire as Supporting Information when you resubmit your manuscript.”

2. Please add a 150- to 200-word non-technical author summary to your article file. The author summary should come after the abstract but before the introduction.

For more information, please see our Author Guidelines:

http://journals.plos.org/globalpublichealth/s/submission-guidelines#loc-author-summary

3. If you have no competing interests to declare, please state "The authors have declared that no competing interests exist". Please update the Competing Interest.

4. Please amend your detailed Financial Disclosure statement. This is published with the article. It must therefore be completed in full sentences and contain the exact wording you wish to be published.

- State the initials, alongside each funding source, of each author to receive each grant.

- State what role the funders took in the study. If the funders had no role in your study, please state: “The funders had no role in study design, data collection and analysis, decision to publish, or preparation of the manuscript.”

5. Please ensure that Funding Information matches the Financial Disclosure Statement.

6. Please provide separate figure files in .tif or .eps format.

7. Please include an editable table and not an image and separate file.

8. Your manuscript is missing the following sections: Introduction. Please ensure these are present, and in the correct order, and that any references to subheadings in your main text are correct. An outline of the required sections can be consulted in our submission guidelines here: 

https://journals.plos.org/globalpublichealth/s/submission-guidelines#loc-parts-of-a-submission

9. We have noticed that you have uploaded Supporting Information files, but you have not included a list of legends. Please add a full list of legends for your Supporting Information files after the references list. 

10. In the online submission form, you indicated that "We aim to put anonymised qualitative data on Bournemouth University data repository [ https://bordar.bournemouth.ac.uk/ ] This process is organisationally time consuming.". All PLOS journals now require all data underlying the findings described in their manuscript to be freely available to other researchers, either 1. In a public repository, 2. Within the manuscript itself, or 3. Uploaded as supplementary information.

Reviewers' comments:

Reviewer's Responses to Questions

**Comments to the Author**

1. Does this manuscript meet PLOS Global Public Health’s publication criteria? Is the manuscript technically sound, and do the data support the conclusions? The manuscript must describe methodologically and ethically rigorous research with conclusions that are appropriately drawn based on the data presented.

Reviewer #1: Partly

2. Has the statistical analysis been performed appropriately and rigorously?

Reviewer #1: N/A

3. Have the authors made all data underlying the findings in their manuscript fully available (please refer to the Data Availability Statement at the start of the manuscript PDF file)?

Reviewer #1: Yes

4. Is the manuscript presented in an intelligible fashion and written in standard English?

Reviewer #1: Yes

5. Review Comments to the Author

Reviewer #1: The manuscript has numerous grammatical errors, which make the point the authours are trying to make murky. The manuscript needs to be submitted for professional language editing.

The authours have made inconclusive statements in some sections, possibly with the assumption that the audience will understand what was meant to be said. This, however, is a mistake as the article may be read and used by audience that are of lower professional knowledge that the audience the authours may have in mind.

6. PLOS authors have the option to publish the peer review history of their article (what does this mean?). If published, this will include your full peer review and any attached files.

**Do you want your identity to be public for this peer review?** For information about this choice, including consent withdrawal, please see our Privacy Policy.

Reviewer #1: No

---

## [Editor Report · Decision Letter 1]

2 Aug 2022

Perceptions around COVID-19 and vaccine hesitancy: A qualitative study in Kaski district, Western Nepal

PGPH-D-22-00044R1

Dear Professor van Teijlingen,

. Y the journal.

Thank you for resubmitting your revised manuscript 'Perceptions around COVID-19 and vaccine hesitancy: A qualitative study in Kaski district, Western Nepal'. You have done a great job of addressing all my suggestions. The manuscript addresses important issues and I hope its publication will help address inequalities. I am very pleased to provisionally accept this manuscript on behalf of PLOS Global Public Health.

Best regards,

Dmitri Nepogodiev

Guest Editor
